Glucosylceramide synthase regulates hepatocyte repair after concanavalin A-induced immune-mediated liver injury

Gan Jian 1
Gao Qin 2
Wang Li Li 3
Tian Ai Ping 4
Zhu Long Dong 4
Zhang Li Ting 4
Zhou Wei 5
Mao Xiao Rong mxr2013@126.com 1 4
Li Jun Feng lijf12@lzu.edu.cn 1 4 5
1 The First Clinical Medical College, Lanzhou University , Lanzhou , Gansu , China
2 Physical Examination Center, The First Hospital of Lanzhou University , Lanzhou , Gansu , China
3 Department of Radiology, The First Hospital of Lanzhou University , Lanzhou , Gansu , China
4 Department of Infectious Diseases, The First Hospital of Lanzhou University , Lanzhou , Gansu , China
5 Institute of Infectious Diseases, The First Hospital of Lanzhou University , Lanzhou , Gansu , China
Uversky Vladimir
Electronic publication date: 2021 Sep 14
Publication date: 2021
Volume: 9
Electronic Location ID: e12138
Received 2021 May 7; Accepted 2021 Aug 18
Copyright: ©2021 Gan et al.
Copyright year: 2021
Copyright holder: Gan et al.
License: This is an open access article distributed under the terms of the Creative Commons Attribution License, which permits unrestricted use, distribution, reproduction and adaptation in any medium and for any purpose provided that it is properly attributed. For attribution, the original author(s), title, publication source (PeerJ) and either DOI or URL of the article must be cited.
License URL: https://creativecommons.org/licenses/by/4.0/

Keywords: Liver injury, Glycosphingolipid, Glucosylceramide synthase, UDP-glucose ceramide glucosyltransferase, Concanavalin A

Funding: National Natural Science Foundation of China 81800528 Natural Science Foundation of Gansu Province 20JR5RA364 Key Research and Development Project of Gansu Province 20YF2FA011 Health Industry Research Project in Gansu Province GSWSKY2018-24 This work was supported by the National Natural Science Foundation of China (No. 81800528); the Natural Science Foundation of Gansu Province (No. 20JR5RA364); the Key Research and Development Project of Gansu Province (No. 20YF2FA011); and the Health Industry Research Project in Gansu Province (No. GSWSKY2018-24). The funders had no role in study design, data collection and analysis, decision to publish, or preparation of the manuscript.

==============================
Background

Sphingolipids produce pleiotropic signaling pathways, and participate in the pathological mechanism of hepatocyte apoptosis and necrosis during liver injury. However, the role of glucosylceramide synthase (GCS)–key enzyme that catalyzes the first glycosylation step, in liver injury is still vague.

Methods

All experiments were conducted using 7–9-week-old pathogen-free male C57BL/6 mice. Serum alanine aminotransferase (ALT) and aspartate aminotransferase (AST) levels were detected in murine models of liver disease, in addition to histological characterization of liver injuries. Quantitative real-time polymerase chain reaction (qRT-PCR) was used to detect the relative expression of the GCS, matrix metallopeptidase-1 (MMP-1), and tissue inhibitor of metalloproteinase-1 (TIMP-1) genes. The GCS was observed through a fluorescence microscope, and the flow cytometry was used to detect hepatocyte apoptosis. The concentrations of serum IL-4, IL-6, and IL-10 were measured using enzyme-linked immune-sorbent assay (ELISA) kit. MMP-1 and TIMP-1 protein expression was measured via western blot (WB) analysis.

Results

Con A is often used as a mitogen to activate T lymphocytes and promote mitosis. A single dose of Con A injected intravenously will cause a rapid increase of ALT and AST, which is accompanied by the release of cytokines that cause injury and necrosis of hepatocytes. In this study, we successfully induced acute immune hepatitis in mice by Con A. Con A administration resulted in GCS upregulation in liver tissues. Moreover, the mice in the Con A group had significantly higher levels of ALT, AST, IL-4, IL-6, IL-10 and increased hepatocyte apoptosis than the control group. In contrast, all of the aforementioned genes were significantly downregulated after the administration of a GCS siRNA or Genz-123346 (i.e., a glucosylceramide synthase inhibitor) to inhibit the GCS gene. Additionally, the histopathological changes observed herein were consistent with our ALT, AST, IL-4, IL-6, and IL-10 expression results. However, unlike this, hepatocyte apoptosis has been further increased on the basis of the Con A group. Moreover, our qRT-PCR and WB results indicated that the expression of MMP-1 in the Con A group was significantly lower than that in the control group, whereas TIMP-1 exhibited the opposite trend. Conversely, MMP-1 expression in the GCS siRNA and Genz-123346 groups was higher than that in the Con A group, whereas TIMP-1 expression was lower.

Conclusions

GCS inhibition reduces Con A-induced immune-mediated liver injury in mice, which may be due to the involvement of GCS in the hepatocyte repair process after injury.

Introduction

MMP-1 and TIMP-1 are closely related to liver fibrosis, however, we do not know their role in early stage of liver fibrosis such as chronic hepatitis or even acute hepatitis. The purpose of this study was to illustrate the relationship between GCS and MMP-1 and TIMP-1, and the influence of MMP-1 on the outcome of acute immune hepatitis. Immune-related chronic liver injuries are among the key reasons for the occurrence and development of multiple liver diseases. For instance, autoimmune hepatitis is a progressive chronic inflammatory liver disease caused by abnormal autoimmune responses and is characterized by a loss of self-tolerance (You et al., 2021). Studies have shown that autoimmune hepatitis has a similar incidence worldwide, with an average of about 1.0 per 100,000. The prevalence depending on the geographical location, specifically, the range in North America is 22.8–42.9 per 100,000, and the range in Europe is 10.7–19.44 per 100,000, which are generally higher than the 12.99 per 100,000 in Asia (Delgado et al., 2013; Hurlburt et al., 2002; Lv et al., 2019). Long-term chronic inflammation and repeated liver injury gradually lead to liver fibrosis, cirrhosis, liver failure, and even death. Liver tissue inflammation mediated by the liver’s immune responses has been linked to the pathological process of viral hepatitis, autoimmune liver disease, liver transplantation, alcoholic liver disease, non-alcoholic fatty liver, and drug-induced liver injury, all of which lead to hepatocyte injury, apoptosis, and necrosis (Heymann & Tacke, 2016; Zhang et al., 2019). Our study found that sphingolipids can activate pleiotropic signaling pathways during immune-mediated liver injury and contribute to hepatocyte injury, apoptosis, and necrosis (Brenner et al., 2013). In fact, our previous studies (Li et al., 2014a; Li et al., 2014b; Li et al., 2017) and other related studies (Grammatikos et al., 2015; Merrill et al., 2009) have demonstrated that sphingolipids and their metabolic enzymes play a key role in the occurrence and development of multiple liver diseases. Glycosphingolipids result from the glycosylation of sphingolipids, and glucosylceramide (GC) is the central molecule in glycosphingolipid metabolism, which is regulated by glucosylceramide synthase (GCS, EC: 2.4.1.80) (Liu, Hill & Li, 2013). GCS, also referred to as UDP-glucose ceramide glucosyltransferase (UGCG), is the key enzyme that catalyzes the first glycosylation step during glycosphingolipid biosynthesis. Therefore, GCS activity directly affects the balance between ceramide and GC in cells, which in turn affects cell survival and immune function and may ultimately lead to disease onset. Based on current studies, we hypothesized that GCS, the key enzyme in glycosphingolipid metabolism, may affect the outcome of immune-mediated liver injury by regulating the hepatocyte repair process after injury. Therefore, an in-depth analysis of the relationship between GCS and immune-mediated liver injury would provide important insights into the mechanisms that regulate the pathogenesis of many immune liver diseases.

Materials & Methods

Ethics statement

This study was approved by the Institutional Animal Care and Use Committee and Scientific Program of Lanzhou University (Ethical Application Ref: LDYYLL2018-100).

Experimental animals

Specific pathogen-free (SPF) male C57BL/6 mice were purchased from the Lanzhou University animal laboratory (qualification certificate number: SCXK2018-0002). These mice were congenic C57BL/6 (backcrossed for five generations) and were then inbred. All mice were maintained in an SPF room (temperature 22–25 °C, 55% humidity, 12-hour light and dark cycle) and fed standard laboratory chow and water ad libitum. All mice used in our experiments were 7–9-weeks-old and were similarly-sized (20 ± 1 g body weight). In order to minimize potential confounders, we chose mice born in the same litter, and the mice used in the experiment had similar body weight.

Reagents

The following reagents were used in our study: concanavalin A (Solarbio Life Science, Beijing); 4% paraformaldehyde (Leagene Biotechnology, Beijing); Genz-123346 (MedChem Express, USA); GCS siRNA: forward 5′-CCC GGU UAC ACC UCA ACA ATT-3′; reverse 5′-UUG UUG AGG UGU AAC CGG GTT-3′ (synthesized by GenePharma Co., Ltd. Shanghai); Entranster™ -in vivo (Engreen Biosystem, Beijing); RNAiso plus, Primescript RT reagent kit with gDNA eraser, TB Green Premix Ex Taq II (Takara Biomedical Technology, Japan); primers for GAPDH, GCS, MMP-1, and TIMP-1 (synthesized by Takara Biomedical Technology, Japan); Mouse IL-4, IL-6, and IL-10 Elisa kits (Elabscience Biotechnology, Wuhan); RIPA lysis solution, BCA Protein Assay Kit, polyvinylidene difluoride (PVDF) membranes, BeyoECL Plus Kit and HRP-labeled secondary antibodies (Beyotime Biotechnology, Shanghai); fluorescent-conjected secondary antibody (Jackson ImmunoResearch Inc., USA); Annexin V-FITC apoptosis kit (BD Pharmingen, 556547, USA); anti-UGCG antibody (BOSTER Biological Technology, BA2382, Wuhan); Cleaved caspase-3 antibody (Cell Signaling Technology, #9664, USA); anti-MMP-1 antibody (Biorbyt Ltd, orb214253, UK); anti-TIMP-1 antibody (Abcam, ab61224, UK).

Experimental mouse models

The mice were allowed to acclimate for 3–7 days prior to the experiments. We selected a small sample size because according to the results of our pre-experiment that 6 mice or even fewer have shown significant statistical differences. Since the mice in the Con A-only group are at risk of death, so the Con A-only groups originally numbered 7 mice. 31 mice were randomly divided into five groups using a computer based random order generator, and the individual mouse was considered the experimental unit within the study. The not treated (NT) group (n = 6) received a single saline injection into the tail vein. The mice in the immune hepatitis group (n = 7) were given a single intravenous injection of Con A (15 mg/kg body weight). The negative control (NC) group (n = 6) not treated with Con A, but with GCS siRNA transfection complex (Construction of transfection complex, siRNA (µg): Entranster™ -in vivo (µl) =2: 1. Mix thoroughly and incubate at room temperature for at least 15 min, then injected into mice via tail vein). The fourth group (n = 6) was treated with a single dose of GCS siRNA transfection complex via tail vein injection two days before Con A administration. The last group (n = 6) received a GCS inhibitor (Genz-123346; 60 mg/kg) twice a day via oral gavage two days before Con A administration. After the Con A injection, the mice were kept in their cages for 24 additional hours. The animals were included in the study if they underwent successful administration of Con A, GCS siRNA or Genz-123346, and the mice were excluded if died prematurely. All experimental procedures were conducted thereafter. In this experiment, we strictly controlled the use of experimental animals, so there were no extra surviving mice. And there were no exclusions except the mice exhibited death, preventing the collection of behavioral, blood and histological data.

Liver enzyme analyses

Thirty minutes before sacrifice, we administered 1% pentobarbital sodium (80 mg/kg, intraperitoneal injection) to the mice. Finally, cervical dislocation was used for euthanizing mice. Blood was obtained by removing the eyeball to analyze serum alanine aminotransferase (ALT) and aspartate aminotransferase (AST) as indicators of hepatocellular injury. Measurements of serum ALT were made using an automatic biochemical analysis instrument (Olympus AU400 series, Tokyo, Japan).

Histological and immunofluorescence

Left outer lobe of liver was excised 24 h after Con A administration, after which the liver samples were fixed in 4% paraformaldehyde and prepared for microscopic assessment using standard procedures. The samples were dehydrated, embedded in paraffin, and cut into 5-µm-thick sections along the long axis of the liver sample (Leica RM2235 Manual Rotary Microtome; Heidelberger, Germany), which were then stained with hematoxylin-eosin, and sections were blindly examined by a pathologist (Dr. Tian). As for immunofluorescence, after deparaffinization and antigen retrieval, 5% goat serum was added to the sections for blocking. The sections were then incubated with GCS-antibody (1:100) overnight at 4 °C, followed by incubation with fluorescent-conjected secondary antibody (1:1000) protect from light for 1 h at 37 °C. 4, 6-diamino-2-phenylindole (DAPI) was used to stain the cell nuclei, and a fluorescence microscope (Nikon-ECLIPSE 80i, Tokyo, Japan) was used to capture the images. Images were acquired as 12-bit file format with both Nikon 4 × and 40 × objectives. Post imaging processing was performed using ImageJ 2×  software (Rawak Software Inc., Stuttgart, Germany).

RNA isolation and quantitative real-time PCR (qRT-PCR) analysis

Total ribonucleic acid (RNA) was extracted from the liver using RNAiso Plus reagent, and RNA integrity was determined by agarose gel electrophoresis. 0.5 to 1.0 µg of total extracted RNA was used to synthesize complementary DNA (cDNA) using Reverse Transcription Kit according to the manufacturer’s instructions. qRT-PCR was performed on a Roche LightCycler480II (Roche Life Science, Basel, Switzerland) using 20-µl reaction volumes (6.4 µl RNase-free water, 2 µl cDNA, 0.8 µl each of forward and reverse primers, and 10 µl TB Green Premix Ex Taq II). qRT-PCR analyses were conducted using the following primers: GAPDH 5′-TGT GTC CGT CGT GGA TCT GA-3′  (forward), 5′-TTG CTG TTG AAG TCG CAG GAG-3′(reverse); GCS 5′-TTC GAG GGC GTG TTA TCC ATC-3′  (forward), 5′-CAA ATG GGC TGG CTC AGT AAG A-3′  (reverse); MMP-1 5′-TCC CTG GAA TTG GCA ACA AAG-3′  (forward), 5′-GCA TGA CTC TCA CAA TGC GAT TAC-3′  (reverse); TIMP-1 5′-GGA ACG GAA ATT TGC ACA TCA G-3′  (forward), 5′-CTG ATC CGT CCA CAA ACA GTG AG-3′  (reverse). Relative gene expression was calculated using the comparative cycle threshold method using GAPDH as reference.

Flow cytometry of apoptosis

In short, the liver was excised and digested by trypsin. Obtain primary hepatocytes and filter repeatedly through cell sieves. The cells were harvested and incubated with reagents from the Annexin V-FITC apoptosis kit according to the manufacturer’s instructions, and the cells were counted using a flow cytometer (Beckman CytoFLEX, California, USA).

Measurement of cytokines in the serum

Serum was collected from individual mice, after which commercially available ELISA kits were used to determine serum IL-4, IL-6, and IL-10 using a multifunctional microplate reader (Thermo Scientific, Massachusetts, USA) according to the manufacturer’s instructions. The mouse IL-4, IL-6, and IL-10 specific polyclonal antibodies were pre-coated onto 96-well plates. The standards, test samples, and biotinylated detection antibodies were then pipetted into the wells, after which unbound biotinylated antibodies were washed away using washing buffer. Biotin-conjugated HRP was pipetted into each microplate well and incubated, then the wells were washed and substrate stop solution was added. Only the wells that contained the corresponding cytokines and biotin-conjugated antibody exhibited a color shift from blue to yellow, after which the optical absorbance at 450 nm was determined using a microplate absorbance reader (Thermo Scientific Varioskan Flash, Massachusetts, USA).

Western blot analysis

Liver tissues were homogenized in ice-cold lysis buffer and incubated for 10 min, then centrifuged (Beckman Coulter life sciences, California, USA) at 12,000× g for 10 min at 4 °C. Protein concentrations were quantified using the BCA Protein Assay Kit. A loading buffer was then added and the proteins were boiled for 5–10 min. Equal amounts of whole protein (30 µg) were separated on 12% acrylamide gels using sodium dodecyl sulfate-polyacrylamide gel electrophoresis. The proteins were then transferred from the gel to PVDF membranes and blocked in TBST containing 5% non-fat milk for 1 h. Next, the membranes were incubated overnight at 4 °C with a specific primary antibody according to the recommended dilution ratio in the instructions. After the incubation period, each PVDF membrane was washed 3–5 times (8 min per wash) with TBST and incubated for 1.5 h at room temperature with HRP-labeled secondary antibodies. The bound antibodies were then visualized using enhanced chemiluminescence detection reagents and exposed using an Amersham Imager 600 system (GE Healthcare, Boston, USA). The signals were analyzed and quantified using the ImageJ 2x software.

Statistical analyses

All data were processed using SPSS 16.0 for Windows software (SPSS, Inc., Chicago, IL), and the results were reported as mean ± standard deviation (SD). The Student t test was used to compare the experimental groups. 95% confidence interval for each statistical analysis and a P-value < 0.05 was considered statistically significant.

Results

General conditions of mice after Con A administration

Prior to the administration of Con A, no significant differences in mental state, behavior, or appearance were observed among the mice regardless of the experimental group. However, the mice treated with Con A alone exhibited malaise, reduced activity, trembling, and huddling to remain warm. Moreover, they responded poorly to external stimuli and their fur became fluffy and messy. The other groups did not exhibit these traits. Moreover, among all of the tested groups, only the Con A-only group exhibited one death. Due to overt behavioral activity the experimenter could not be blinded to whether the animal was injected with Con A.

GCS inhibition reduces Con A-induced hepatic enzyme upregulation

As shown in Figs. 1A and 1B, serum ALT and AST levels were significantly elevated in the Con A group compared to those in the NT group. Moreover, we noticed that the GCS mRNA expression of the Con A-exposed mice was significantly higher than that of the NT group (Fig. 1C). We then treated the mice with GCS siRNA and Genz-123346 to suppress GCS expression. As expected, serum ALT and AST levels were significantly lower in the mice that were pre-treated with GCS siRNA and Genz-123346 than in the mice that were exposed to Con A alone.

Figure 1 Serum ALT, AST levels and relative expression of GCS mRNA.

(A, B) Serum ALT and AST levels. (C) The relative expression of GCS mRNA. (D) 1.2% Tris, borate and ethylenediamine tetraacetic acid (TBE) agarose gel electrophoresis of representative RNA samples.

GCS inhibition ameliorates Con A-induced liver injury

Based on our histological analyses of liver samples, the mice in the Con A group (Fig. 2B) exhibited areas with widespread necrosis and extensive infiltration of inflammatory cells around the central veins (i.e., clear signs of inflammation) compared to the NT group (Fig. 2A). In contrast, the liver tissue sections from the mice that were pre-treated with GCS siRNA (Fig. 2C) and Genz-123346 (Fig. 2D) only exhibited some isolated areas of necrotic tissue, indicating that the majority of the hepatocytes were less sensitive to Con A-induced liver injury. By immunofluorescence (Fig. 3), we observed the GCS level was increased in the Con A group. As expected, the GCS siRNA+Con A and Genz+Con A groups showed a decrease GCS expression compared to the Con A group.

Figure 2 Photomicrographs of representative sections of liver at 24 h after Con A injection.

(A and F) Normal liver section, no necrotic areas. (B and G) Widespread necrotic areas and inflammatory cells infiltrate, mainly lymphocytes. (C and H) no necrotic areas. (D and I) Some isolated necrotic areas and lymphocytes infiltrate. (E and J) Some isolated necrotic areas and lymphocytes infiltrate. A–E represents the NT group, Con A group, NC group, GCS siRNA+Con A group and Genz+Con A group in turn, H&E, 4 ×. F–J represents the NT group, Con A group, NC group, GCS siRNA+Con A group and Genz+Con A group in turn, H&E, 40 ×. Red wireframe represents the necrotic areas, and black arrows indicate the central vein of hepatic lobules.

Figure 3 Expression of GCS protein in liver.

(A) The representative immunofluorescent images of GCS. DAPI labeled nucleus (blue); DyLight 594 (red) labeled GCS. Scale bars are 50 µm. (B) GCS protein were analyzed by western blot. (C) Relative levels of GCS protein corrected by GAPDH.

Increased hepatocytes apoptosis rate after inhibiting GCS

Flow cytometry analysis (Figs. 4A and 4B) revealed that the GCS siRNA+Con A and Genz+Con A group showed more apoptosis rate compared to the Con A group, especially the Genz+Con A group, with an apoptosis rate of 21.6% compared with the other three groups 1.36%, 7.05%, 12.5%, respectively. Similarly, inhibition of GCS in normal mice (NC group) caused hepatocyte apoptosis, but the apoptosis rate was not as obvious as that in Con A-treated mice.

Figure 4 Apoptosis rate of hepatocyte.

(A) Flow cytometric analysis of cellular apoptosis using annexin V/PI double staining. (B) Statistical analysis of hepatocyte apoptosis rate. (C) Cleaved caspase-3 activity were analyzed by western blot. (D) Relative levels of cleaved caspase-3 activity corrected by GAPDH.

Inhibiting GCS alleviates the release of inflammatory cytokines caused by Con A-induced liver immune responses

Con A-induced liver injuries provide a robust basis for the establishment of immune liver injury models, as this compound can activate T lymphocytes and promotes the release of large amounts of cytokines into the bloodstream. Next, to investigate the role of GCS in Con A-induced autoimmune hepatitis, we quantified the levels of multiple cytokines in the serum of mice. Con A-injected mice exhibited significantly higher IL-4, IL-6, and IL-10 serum levels (Figs. 5A, 5B, and 5C, respectively) compared to the NT group. As expected, after GCS suppression, the mice in the GCS siRNA+Con A and Genz+Con A groups had significantly lower IL-4, IL-6, and IL-10 serum levels compared to the Con A group.

Figure 5 Serum IL-4, IL-6 and IL-10 levels.

(A) Serum IL-4 level. (B) Serum IL-6 level. (C) Serum IL-10 level.

GCS affects the hepatocyte repair process by regulating MMP-1 and TIMP-1

Next, the levels of MMP-1 (Figs. 6A and 6E) and TIMP-1 (Figs. 6B and 6F) were quantified to explore whether the mitigation of immune-mediated liver injury by GCS was due to an enhanced hepatocyte repair function. Compared with the NT group, the TIMP-1 levels in the liver were significantly higher after con A injection, whereas MMP-1 exhibited the opposite trend. Conversely, hepatic MMP-1 levels increased compared with the Con A group, whereas TIMP-1 decreased gradually. However, inhibition of GCS in normal mice (NC group) have no obvious effect on the levels of MMP-1 and TIMP-1.

Figure 6 MMP-1 and TIMP-1 levels in liver.

(A) The relative expression level of MMP-1 mRNA measured by qRT-PCR method. (B) The relative expression level of TIMP-1 mRNA measured by qRT-PCR method. (C and D) MMP-1 and TIMP-1 proteins were analyzed by western blot. (E) Relative levels of MMP-1 protein corrected by GAPDH. (F) Relative levels of TIMP-1 protein corrected by GAPDH.

Discussion

Increasing evidence suggests a link between GCS expression and disease onset, thus highlighting the value of GCS as a promising therapeutic target (Pavlova et al., 2015; Wang et al., 2015; Wegner et al., 2018). Here, we demonstrated that the GCS played a key role in immune hepatitis. GCS inhibition had beneficial effects in murine models of Con A-induced hepatitis, and these reductions in liver injuries appeared to be related to the regulation of hepatocyte repair function by GCS.

The liver is not only a major metabolic organ but also an important immune system organ. Moreover, its various constituent cells (e.g., T lymphocytes, B lymphocytes, natural killer (NK) cells, and natural killer T (NKT) cells) are involved in several immune responses (Gan et al., 2020; Zhang et al., 2019). Physical and chemical factors, viruses, drugs, and alcohol can lead to chronic hepatocyte injury, apoptosis, and necrosis via the activation of immune responses, thus resulting in liver diseases such as viral hepatitis, drug-induced liver injury, autoimmune liver disease, and alcoholic liver disease (Smyk et al., 2018; Zhang et al., 2019). Therefore, immune-related chronic liver injury has recently been linked to the occurrence and development of multiple liver diseases. T lymphocyte-mediated cellular immune responses are generally known to be the main factor that leads to hepatocyte injury in immune hepatitis (Loggi et al., 2014). This is especially true for natural killer T cells, which are abundant in the liver, particularly in rodents (Opasawatchai & Matangkasombut, 2015). NKT cells secrete high levels of cytokines (including Th1 and Th2) when stimulated by antigenic substances, thereby participating in the regulation of innate and adaptive immunity and affecting the outcome of immune-mediated liver injury (Popovic et al., 2017). Recent studies have demonstrated that sphingolipids play a key role in regulating the development, maturation, and homeostasis of NKT cells in the liver. Therefore, mice lacking enzymes for sphingolipid synthesis exhibit reduced immunity and are susceptible to pathogen infection (Saroha et al., 2017). GCS is a key enzyme for the synthesis of glycosylated sphingolipids, as this enzyme can generate all complex glycosyl phospholipids precursors, including glucosylceramide (GC). GCS is a key node for lipid glycosylation and non-glycosylation regulation, and thus its activity directly affects the metabolic balance of glycosylated and non-glycosylated sphingolipids, which in turn affects cell survival and immune function (Popovic et al., 2017). Therefore, GCS plays a key role in immune hepatitis and affects its outcome. Con A is a kind of plant lectin extracted from jack beans (Canavalia ensiformis). This compound binds to the sugar chain residues on the cell surface to promote cell aggregation and T lymphocyte mitosis, thus inducing lymphocyte and macrophage toxicity (Wang et al., 2017). Therefore, Con A-induced liver injury is typically used as a model to simulate human immune hepatitis (Li et al., 2015). In this study, we constructed a liver injury model by injecting Con A (15 mg/kg) into the tail vein of mice, after which the serum levels of ALT and AST increased rapidly. ALT is ubiquitous in the body; however, its highest concentrations are typically observed in the cytoplasm and mitochondria of hepatocytes, which makes this enzyme a good indicator for hepatocyte injury diagnosis (Fullerton, Roth & Ganey, 2013; Tsutsui & Nishiguchi, 2014). Moreover, Con A-treated mice also released a large amount of inflammatory cytokines (IL-4, IL-6, and IL-10) into their bloodstream, and histopathological examination revealed observable liver injuries and necrosis. In short, all of the aforementioned reports are consistent with the symptoms of immune hepatitis, including inflammatory cell infiltration, hepatocyte necrosis, and the release of a large number of inflammatory factors (Wang et al., 2015). Genz-123346 is a selective inhibitor of GCS, which specifically inhibit GCS, whether in vivo (Natoli et al., 2010) or in vitro (Chai et al., 2011). The use of Genz-123346 will significantly inhibit the activity of GCS and reduce the level of GC, which leads to an increase in the level of ceramide (the precursor of glucosylceramide), but it didn’t significantly alter the levels of other sphingolipids such as sphingosine 1-phosphate, sphingomyelin or sphingosine (Koike et al., 2019). Afterward, we inhibited the activity of GCS by administering effective doses of GCS siRNA and Genz-123346 (i.e., a GCS chemical inhibitor). As expected, the levels of serum ALT, AST, and inflammatory cytokines (IL-4, IL-6, and IL-10) in mice were significantly reduced, and liver histopathological analyses revealed a substantial reduction in liver injuries. Interestingly, as GCS is inhibited, the rate of hepatocyte apoptosis is increasing. This may be that after GCS is inhibited, the conversion of ceramide into GC is reduced, and a large amount of ceramide accumulates in hepatocyte and causing apoptosis (Stefanovic et al., 2016). In addition, GCS may be the necessary protein for the proliferation, since the GCS knockout mice exhibit embryonic lethality due to massive apoptosis (Yamashita et al., 1999). Furthermore, this may be a self-protection mechanism, because the liver resists severe injury by increasing the number of apoptosis. These results suggest that GCS inhibition can reduce Con A-induced liver injury in mice and inhibits the release of inflammatory cytokines.

MMP-1 is mainly produced by hepatic stellate cells and is a type of Zn2+- and Ca2+-dependent endogenous protease that exists in zymogen form. It can degrade a variety of extracellular matrix (ECM) components to reduce collagen deposition, and play a critical role in alleviating liver fibrosis. Excessive ECM deposition is known to cause organ fibrosis (e.g., liver fibrosis) and therefore the relationship between MMP-1 and liver fibrosis has recently garnered increasing attention from the scientific community. Liver fibrosis is a common terminal manifestation of chronic liver disease characterized by persistent liver parenchymal injury, activation and recruitment of immune cells, activation of HSC, and excessive secretion of ECM, all of which lead to scar formation (Roderfeld, 2018). Therefore, the liver appears to undergo a long inflammatory response period before the development of liver fibrosis, during which inflammatory injuries are repeatedly repaired. If this inflammatory response is not effectively controlled, chronic liver disease ultimately develops into liver fibrosis. Previous studies have demonstrated that MMP-1 is not only an important indicator of ECM degradation during liver fibrosis, but also maintains the integrity of the cell structure and the basement membrane of the liver sinus during the repair process after injury, which occurs prior to the development of liver fibrosis (Roderfeld, 2018). Therefore, MMP-1 may play a key role in hepatitis onset and can thus be used as an indicator of cell repair capacity. Tissue inhibitor of metalloproteinase-1 (TIMP-1) has a specific inhibitory effect on MMP-1, which reduces the degradation of ECM by MMP-1 and leads to the development of liver fibrosis. Therefore, it is believed that the dynamic balance between MMP-1 and TIMP-1 plays a decisive role in the outcome of liver injury. In order to study how GCS can reduce hepatocyte injury in Con A-induced immune hepatitis, we chose to explore MMP-1 and TIMP-1. In this study, the experimental mice exhibited obvious signs of liver injury with a significant increase in inflammatory cytokines after Con A administration, all of which coincided with a substantial upregulation of the GCS gene. Furthermore, both our qRT-PCR and WB results supported the decrease in MMP-1 and increase in TIMP-1 expression in the livers of the Con A group mice. Interestingly, the opposite occurred after GCS inhibition, resulting in a marked decrease in liver injuries. During liver injury, the upregulation of TIMP-1 inhibits MMP-1 activity, resulting in a decrease in the repair capacity of hepatocytes, and therefore the integrity of the hepatocyte structure and the normal morphology of the basement membrane of the liver sinus cannot be maintained. Liver injuries then become progressively more severe and gradually develop into liver fibrosis (Bataller & Brenner, 2005). Conversely, a decrease in TIMP-1 leads to a weakened inhibition of MMP-1, which protects the hepatocytes. Here, we provide further evidence that inhibiting GCS can increase MMP-1 expression, thereby enhancing hepatocyte repair.

There are several potential limitations of this study that need to be discussed. First, although our preliminary findings suggest that GCS affects the outcome of immune liver injury by regulating the levels of MMP-1 and TIMP-1 in liver tissue, we do not know whether GCS regulates other types of MMPs and TIMPs, and future studies should also determine whether there are differences between MMP and TIMP subtype functions and if there are synergistic or antagonistic interactions between them. In addition, this study is not thorough enough on the changes in T lymphocytes and their subtypes in Con A induced immune hepatitis. This will be the focus of our future work, and we are already working on it.

Conclusions

Con A-induced organ injury provides a simple means to recreate liver-specific injuries and does not require pre-sensitization (Heymann et al., 2015). Using this approach to create a murine model of immune-mediated liver injury, our study demonstrated the critical role of GCS in liver disease onset. GCS inhibition enhances the self-repair capacity of hepatocytes, thus alleviating disease symptoms. Moreover, our findings suggest that regulating MMP-1 through different means may serve as a promising treatment for liver injury; however, these findings must be further confirmed in other models.

Supplemental Information

Supplemental Information 1 Raw data for Figs. 1D, 2A and 2F

Click here for additional data file.

Supplemental Information 2 Raw data for Figs. 2B and 2G

Click here for additional data file.

Supplemental Information 3 Raw data for Figs. 2C and 2H

Click here for additional data file.

Supplemental Information 4 Raw data for Figs. 2D and 2I

Click here for additional data file.

Supplemental Information 5 Raw data for Figs. 2E and 2J, 3A and 3B, 4C, 6C, and 6D

Click here for additional data file.

Supplemental Information 6 ARRIVE 2.0 checklist

Click here for additional data file.

Supplemental Information 7 Raw numerical data for all statistical analysis

Click here for additional data file.

We are grateful to the Central Laboratory of the First Hospital of Lanzhou University for providing us with experimental support. In addition, we are grateful to all the staff in this study.

Additional Information and Declarations

Competing Interests

Author Contributions

Animal Ethics

Data Availability

The authors declare there are no competing interests.

Jian Gan performed the experiments, analyzed the data, prepared figures and/or tables, authored or reviewed drafts of the paper, and approved the final draft.

Qin Gao performed the experiments, analyzed the data, prepared figures and/or tables, and approved the final draft.

Li Li Wang and Wei Zhou performed the experiments, prepared figures and/or tables, and approved the final draft.

Ai Ping Tian and Long Dong Zhu analyzed the data, prepared figures and/or tables, and approved the final draft.

Li Ting Zhang conceived and designed the experiments, prepared figures and/or tables, and approved the final draft.

Xiao Rong Mao and Jun Feng Li conceived and designed the experiments, authored or reviewed drafts of the paper, and approved the final draft.

The following information was supplied relating to ethical approvals (i.e., approving body and any reference numbers):

This study was approved by the Institutional Animal Care and Use Committee and Scientific Program of Lanzhou University (Ethical Application Ref: LDYYLL2018-100).

The following information was supplied regarding data availability:

The raw measurements are available in the Supplemental Files. These original data mainly include original, unedited images for all blots, uncropped scans or photographs of the histological and immunofluorescence images.

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
