# Peer review of "Glucosylceramide synthase regulates hepatocyte repair after concanavalin A-induced immune-mediated liver injury"

_PeerJ, doi:10.7717/peerj.12138_

## Round 0.1 · original submission · Major Revisions

1) It is not clear in the manuscript if the inhibition of Glucosylceramide synthase (GCS) with the siRNA and drug inhibitor is systemic or liver-specific, due to the fact that they are administered via tail vein and that GCS is expressed in other tissues such as kidney and tonsil.

2) Control groups not treated with Con A, but with siRNA GCS or inhibitor would be necessary.

3) As raised by reviewer No.1, a control with siRNA scramble would seem indispensable.

4) According to immunofluorescence, the siRNA used in the manuscript does not inhibit GCS expression at protein level. siRNA and inhibitor groups have higher expression levels than the control group. This contrasts with the qRT-PCR results. However due to the fact that the siRNA is not capable of counteracting the effects of Con A on GCS expression, its use in this manuscript seems dispensable. The results of the inhibitor are more consistent.

5) An evaluation of the activity of GCS in this manuscript would be very useful. Perhaps it could be measured by the levels of glucosylceramide, which is the product of the reaction catalyzed by GCS. This in light of the fact that the siRNA does not seem to affect GCS expression effectively in the presence of Con A.

6) It is not clear what is the purpose of figure 1D. It appears to show RNA integrity, however the molecular weight marker is missing and it is not mentioned in the text.

7) The fact that the drug inhibitor decreases GCS expression is outstanding. Are the previous reports of this? Is the mechanism known? It is worth mentioning that in the discussion.

8) Figure 2 should include arrows or by other means indicate the necrotic areas, blood vessels and immune cells present in the micrographs. It would be worth describing what section of the liver is being analyzed, how was the liver sectioned. The micrograph 2B seems blurry and should be substituted for a better one. The use of immunohistochemical markers,like the ones suggested by Reviewer No.2 would help to clarify the impact of GCS inhibition on immune infiltration.

9) GCS inhibition reduces necrosis (figure 2) but increases apoptosis, an explanation of this should be included in the discussion. Induction of apoptosis seems to contradict the beneficial effects of GCS inhibition on liver injury.

10) The molecular weight of MMP-1 is 53kDa, but in figure 6 it says that it is 23kDa

11) The number of experimental units in each figure should be noted, it is not clear if for every experiment samples from all 6 animals per group were analyzed. Figure 1D that has 3 lanes for each experimental group incites this doubt.

12) Reviewer No.2 noted various language and figure style corrections that should be addressed, see his comments for better reference.

Reviewer 1 ·

Basic reporting

The manuscript “Glucosylceramide synthase regulates hepatocyte repair after concanavalin A-induced immune-mediated liver injury” by Jian Gan and collaborators was critically reviewed.

The manuscript is aimed to figure out the relevance of GCS in Con A-induced immune-mediated liver injury in mice.

Although the work presents interesting data, some concerns should be addressed.

Experimental design

Some control group is required, please see below.

Validity of the findings

Please see below.

Additional comments

The manuscript “Glucosylceramide synthase regulates hepatocyte repair after concanavalin A-induced immune-mediated liver injury” by Jian Gan and collaborators was critically reviewed.

The manuscript is aimed to figure out the relevance of GCS in Con A-induced immune-mediated liver injury in mice.

Although the work presents interesting data, some concerns should be addressed.

The authors must provide convincing data demonstrating that animals under the GCS siRNA transfection indeed decreased the GCS protein content and activity; GCS mRNA quantification is not enough.
A complete technical description of the procedure must be provided, including the siRNA sequence and the method for animal assimilation, particularly in the liver.
How was the siRNA transfected into liver cells? Please explain this aspect clearly in the article.

Some reporter gene or siRNA control should be used to demonstrate the efficiency of the transfection. Even more, it is mandatory to include an animal group subjected to a mock siRNA.
Figure 1D must include the reference (standard ruler marker to identify the target band); otherwise, the experiment has no sense.

Because the entire work is based on this animal treated with the siRNA, all the results are irrelevant until these controls and evidence that the experiment using siRNAs was conducted efficiently.

Findings in histology (H&E staining) must be indicated; for example, use arrows to show the necrotic areas. It would be beneficial to suggest inflammatory cell infiltration in the tissue to support the immune system participation. Please provide some greater magnification to observe changes in morphology. Please indicate some reference, for example central vein (CV) or Portal vein (PV).

Figure 3 depicts GCS determination by IF, but this should be supported by western blot of the GCS. The images are not clear some doubts remain because it could be confused by background fluorescence.

Apoptosis should be verified by an alternative method, for example Caspase 3 activity.

Western blot in Figure 6C must include at least two samples of each animal group.
Animal group label as control should be referred to as Not treated (NT) because other control groups are in the study.

Reviewer 2 ·

Basic reporting

The experimental study by Jian-Gan et al. has been designed to investigate the action of Glucosylceramide synthase (GSC), in the regulation of hepatocyte repair after concanavalin A-induced immune-mediated liver injury, using 7-9-week-old pathogen-free male C57BL/6 mice. The authors performed an experimental design by GCS siRNA transfection before Con A administration. Authors report that administration of GSC in mice on Con A injection downregulated the hepatic expression of IL-4, IL-6, and IL-10. Moreover, the authors also report that administration of GCS reduced ALT, AST levels. Authors conclude that GCS inhibition reduces Con A-induced immune-mediated liver injury in mice, which may be due to the involvement of GCS in the hepatocyte repair process after injury. In addition, I consider that it should be improved the clarity and professional language of writing.

I suggested including “Con A” meaning in results from the abstract. Because of up to lines 98 and 283-284, it mentioned, “concanavalin A” and “Con A is a kind of plant lectin extracted from jack beans (Canavalia ensiformis)”.

The introduction provides a worthy, generalized background of the topic that gives the reader an appreciation of the importance of study the molecular mechanism of glucosylceramide synthase in glycosylation metabolism by regulating the hepatocyte repair process after injury. Although to make the introduction more considerable, the author may wish to provide percentages or data about immune-related chronic liver injuries, specifically autoimmune hepatitis prevalence or incidence to substantiate the claim made in the 60-65 lines. Besides updated references about this suggestion.

Articles proposal:
1. Tunio NA, Mansoor E, Sheriff MZ, Cooper GS, Sclair SN, Cohen SM. Epidemiology of Autoimmune Hepatitis (AIH) in the United States Between 2014 and 2019: A Population-based National Study. J Clin Gastroenterol. 2020 Oct.
2. Sucher E, Sucher R, Gradistanac T, Brandacher G, Schneeberger S, Berg T. Autoimmune Hepatitis-Immunologically Triggered Liver Pathogenesis-Diagnostic and Therapeutic Strategies. J Immunol Res. 2019 Nov 25;2019:9437043.
3. Lv T, Li M, Zeng N, Zhang J, Li S, Chen S, Zhang C, Shan S, Duan W, Wang Q, Wu S, You H, Ou X, Ma H, Zhang D, Kong Y, Jia J. Systematic review and meta-analysis on the incidence and prevalence of autoimmune hepatitis in Asian, European, and American population. J Gastroenterol Hepatol. 2019 Oct;34(10):1676-1684.

Experimental design

The hypothesis is clearly defined in 78-80 lines. However, I suggested adding an aim “The purpose of this study was to…” in the introduction in line 60 line to provides more clarity and comprehension for the reader since the beginning of the article.

In Figure 2, the tissue picture of HE subsection B) is of poor quality. A better picture should be presented. Another suggestion is to include labels in the figure to improve clarity ( A-D represents the control group, Con A group, GCS siRNA+Con A group, and Genz + Con A group).

The experimental study by Jian-Gan et al. is novel and of potential interest in the field of immune hepatitis. Data and conclusions reported are convincing and I have just a minor request:

Authors should offer, morphological evidence that inflammatory infiltrate is increased in the Co A group compared to the Control group (immune-histochemistry for CD68, F4/80, for example).

Furthermore, after stating that the choice of flow cytometry technique for evaluating apoptosis rate is important, the author offers a clear explanation of why he chooses this method in the present work. I think the motivation for the present research would be stronger if the author could provide a more direct link between the importance of choosing an appropriate method and the results obtained.

In material and methods, I consider including the name of the image analysis program they used and the number of images captured in histological and immunofluorescent in line 146.
In line 150, there is an error of "electrophoresi" word, inside of electrophoresis.
I recommend improving the English language and redaction of 148-152 lines and 174-176 lines for more clarity and understanding from readers.

Validity of the findings

As suggested above, I think a more in-depth discussion of Figure 4 would be helpful. I consider this is an important result for this article, and therefore it merits more discussion.

No significant limitations are discussed. It may be worthwhile to mention the tradeoffs involved in choosing flow cytometry as opposed to some other method.

Additional comments

Did a pathologist confirm the histological analysis of the HE stain?
I recommend include arrows in figure 2 and figure 3, that represents explication in results.
More separate scale in figure 6B (example 0,1,2,3,4). Homogenize graphics formats in "y axis".

---

## Round 0.2 · accepted · Accept

The original Academic Editor is unavailable and so I have stepped in as Section Editor to make this decsiion.

All critiques were adequately addressed and the manuscript was amended accordingly. Therefore I am pleased to accept your manuscript in its present form.

Reviewer 1 ·

Basic reporting

The manuscript has been significantly improved.

Experimental design

The experimental design is correct.

Validity of the findings

Main findings have been effectively validated in the current revision version

Additional comments

The authors have responded to all reviewers´ comments. I think it is suitable for its publication in its current form

Reviewer 2 ·

Basic reporting

Authors have complied with my queries.
No comment.

Experimental design

Authors have complied with my queries.
No comment.

Validity of the findings

Authors have complied with my queries. No comments.

Additional comments

Authors have complied with my queries. No comments.